# A Hybrid System for Defect Detection on Rail Lines through the Fusion of Object and Context Information

**DOI:** 10.3390/s24041171

**Published:** 2024-02-10

**Authors:** Alexey Zhukov, Alain Rivero, Jenny Benois-Pineau, Akka Zemmari, Mohamed Mosbah

**Affiliations:** 1Univ. Bordeaux, CNRS, Bordeaux INP, INRIA, LaBRI, UMR 5800, 33400 Talence, France; jenny.benois-pineau@u-bordeaux.fr (J.B.-P.); akka.zemmari@u-bordeaux.fr (A.Z.); mohamed.mosbah@u-bordeaux.fr (M.M.); 2Ferrocampus, 18 bd Guillet-Maillet, 17100 Saintes, France; 3SNCF-Réseau, Direction Générale Industrielle et Ingénierie, Département IP3M DM MATRICE, 6 Avenue François Mitterrand, 93574 Paris, France; alain.rivero@reseau.sncf.fr

**Keywords:** image sensors, object recognition, fusion, attention models

## Abstract

Defect detection on rail lines is essential for ensuring safe and efficient transportation. Current image analysis methods with deep neural networks (DNNs) for defect detection often focus on the defects themselves while ignoring the related context. In this work, we propose a fusion model that combines both a targeted defect search and a context analysis, which is seen as a multimodal fusion task. Our model performs rule-based decision-level fusion, merging the confidence scores of multiple individual models to classify rail-line defects. We call the model “hybrid” in the sense that it is composed of supervised learning components and rule-based fusion. We first propose an improvement to existing vision-based defect detection methods by incorporating a convolutional block attention module (CBAM) in the you only look once (YOLO) versions 5 (YOLOv5) and 8 (YOLOv8) architectures for the detection of defects and contextual image elements. This attention module is applied at different detection scales. The domain-knowledge rules are applied to fuse the detection results. Our method demonstrates improvements over baseline models in vision-based defect detection. The model is open for the integration of modalities other than an image, e.g., sound and accelerometer data.

## 1. Introduction

Rail networks play an important role in transportation around the world. Defect detection on rail lines is a crucial aspect of ensuring safe and efficient transportation. The continuous use of rail tracks can lead to various defects, such as cracks, wear, and corrosion. If the defects remain unnoticed and are not regularly inspected, they might pose a significant threat to the safety of passengers and cargo.

In order to prevent accidents and ensure the smooth operation of trains, various methods of defect detection are employed. These methods range from visual inspections [1] to such technologies as ultrasonic [2] and electromagnetic testing [3] and are a part of a vast problematic of defect detection [4]. With the increasing demand for faster and more efficient rail transportation, the traditional methods are often unable to deliver accurate results while the train is running on the tracks. Hence, in order to speed up the process and increase accuracy when constantly evaluating rail lines for defect detection, various deep learning paradigms can be used.

Our research is conducted in the framework of a large research and innovation project conducted by the French Railway company (SNCF): the “Innovative Light Train”. This project aims to revitalize rural rail lines with cutting-edge technologies. One of these technologies is the object of this research: to use artificial intelligence to detect rail defects based on the rail images, domain knowledge, and other information coming from different sensors, such as sound sensors (microphones) and dynamic sensors (accelerometers).

Nevertheless, the prevalent modality for defect detection has always been the imaging modality [5]. The majority of the methods solely focus on the defect itself [6]. This is partially due to the willingness to imitate human perception of visual scenes. Humans focus on semantic objects, and this helps in designing more efficient object-recognition approaches [7]. In the case of specific defects which might be considered “objects difficult to detect”, the knowledge of co-occurrence of contextual objects is beneficial. Thus, one has to build a detector architecture which allows for the fusion of object and context information. The architecture has to be extendable as different modalities might need to be integrated. Indeed, the mixing of modalities in artificial intelligence (AI)-based decision schemes [8] strongly improves accuracies.

The contribution of our research consists of the following:A proposal of an efficient and extensible fusion scheme for detection of defects with both object-of-interest and contextual information integration;Detectors of objects and context, both built on YOLOv5 [9] models, with improved performances by integrating attention mechanisms.

The remainder of this paper is organized as follows. In Section 2, we present a state-of-the-art (SOTA) analysis on defect detection and the AI approaches needed. Section 3 presents the architecture of our system and details different solutions proposed for its components. In Section 4, we present the taxonomy of defects and also the domain knowledge describing the co-occurrence of defects and contextual elements with experiments. Results are discussed in Section 5. Finally, Section 6 concludes this work and outlines its perspectives.

## 2. Related Works

This section presents various techniques such as vision-based defect detection methods, attention modules in DNNs, and different types of data fusion techniques.

Following the publication of the groundbreaking paper [10], deep learning techniques gained widespread popularity in various sectors, including railways. Convolutional neural networks (CNNs) have demonstrated superior performance over traditional machine learning algorithms in numerous practical scenarios involving vision-based methods. A key benefit of deep learning methods is their adaptability to different domains without the need for modifications of the algorithms.

### 2.1. Vision-Based Defect Detection

Vision-based defect detection systems are automated inspection systems that use computer vision to identify defects, anomalies, or irregularities on manufactured products or on material surfaces. These systems are widely used in manufacturing, electronics, automotive, and other industries to ensure product quality and reduce waste.

In recent decades, there has been a growing interest in leveraging deep learning to address various challenges of computer vision. These challenges encompass a wide range of tasks, such as object detection [11], object tracking [12], image classification [13], and semantic segmentation [14].

Concerning defect detection, the primary emphasis is on object detection, as defects are treated as entities requiring both localization and classification. Deep learning-based defect [15] detection algorithms prioritize data-driven feature extraction. By leveraging vast datasets, these algorithms extract deep features, offering distinct advantages over surface-level defect detection methods. In the realm of deep learning, vision-based defect inspection algorithms can broadly be categorized into two classes: classification-based methods and regression-based methods, which are as follows:**Classification-based defect detection methods** consist of various algorithms represented by region-based convolutional neural network (R-CNN) series including R-CNN [16], SPP-Net [17], fast R-CNN [18], region-based full CNN [19], and mask R-CNN [20]. Based on the following algorithms, Fan et al. [21], Ji et al. [22], Zhang et al. [23], Guo et al. [24], Jin et al. [25], and Cai et al. [26] inspected surface defects of wood, gear, metal, etc. These algorithms have a two-stage processing method. Initially, the given image is divided into region proposals that might contain objects. Using a pretrained CNN, these proposed regions are converted into feature vectors. Later, these feature vectors are used for object classification. R-CNN algorithms are slow due to the need to process each proposed region separately. To achieve high accuracy, R-CNN algorithms usually require high computing power because of their complex model architecture. These methods have a relatively low inference speed on lower graphics processing units (GPUs) compared to regression-based methods.**Regression-based object detection methods** are a class of object detection techniques that use regression models to predict the bounding box coordinates and class probabilities for objects in an image. These methods typically involve training a CNN to learn the relationship between input images and their corresponding object bounding boxes and class labels. Regression-based algorithms are characterized by only one round of processing, so the inference speed is faster even on lower GPUs. Redmon et al. [27] proposed the well-known YOLO algorithm, which is a representative regression-based and end-to-end model. Moreover, the other regression-based algorithms are SSD [28], CornerNet [29], and EfficientDet [30]. YOLOv3 [31], YOLOv5, and YOLOv8 [32] are among the most widely used YOLO algorithms for object detection. Based on YOLOv3, Jing et al. [33], Li et al. [34], Huang et al. [35], and Du et al. [36] performed surface defect inspections of fabric, PCB boards, pavements, etc. Although there have been proposed later versions of YOLO detectors, YOLOv5 is a reasonable object detection algorithm which has incorporated a multiscale approach to tackle defects of various sizes. A detailed description of YOLOv5 is provided in Section 3.2. Nevertheless, a comparative analysis between YOLOv5 and YOLOv8 is provided in Section 5.1.

### 2.2. Fusion Methods

Current commercial systems using convolutional neural networks show that image analysis by CNNs is a difficult task due to the ambiguity of the interpretation of the information contained in the images. The detected elements have strong contextual dependencies due to their spatial interactions: joints are assimilated to breaks (Figure 1a), holes in the central part of the rail are confused with missing nuts, and surface defects are often confused with vegetation or oil stains (Figure 1b). In this context, the implementation of a fusion model constitutes a simple and effective tool to solve these problems [37]. Furthermore, it provides an appropriate theoretical framework for dealing with these data and their imperfections. Thus, the presence of a fishplate makes it possible to remove ambiguity when detecting rail breaks and missing nuts.

In defect detection problems, it is interesting to fuse information coming from different sources. The well-known fusion modes are as follows: (i) early fusion in the data space, (ii) intermediate fusion in the learned representation space, and (iii) late fusion in the decision space. When domain knowledge has to be incorporated in the fusion approach, it is more appropriate to consider the third mode, late fusion, as it allows for an easy incorporation of domain knowledge in the decision-making process.

Late fusion is a data integration technique where data sources are utilized independently and then combined at a later stage during decision making, as shown in Figure 2. It draws inspiration from ensemble classifiers’ popularity [38]. Late fusion offers a straightforward alternative to early fusion, especially when the data sources differ significantly in terms of sampling rate, data dimensionality, and units of measurement. This method often yields improved performance because errors from multiple models are handled independently, thereby reducing correlation among errors. However, Ramachandram et al. [39] argue that there is no conclusive evidence supporting the superiority of late fusion over early fusion. Nevertheless, many researchers employ late or decision-level fusion in multimodal data problems [40,41,42]. Various rules are available to determine the optimal approach for combining independently trained models. Some commonly used late-fusion rules include Bayes rules, max fusion, average fusion, and learned-score fusion [43]. In the next section, we will present our hybrid approach for rail defect detection based on deep learning and domain-knowledge late fusion.

## 3. Hybrid Rule- and DNN-Based Defect Detection Architecture

In this section, we present the overall system of defect detection on the rails. We call it “hybrid” in the sense that it uses both rules from domain knowledge on the co-occurrence of specific defects and context elements and a deep learning approach. The backbones of deep learning approaches are YOLO detectors; in particular, we use YOLOv5 [44] and improve it by the inclusion of the attention module CBAM [45]. Nevertheless, we stress that this version of YOLO detectors can be substituted by any other later version and also other object detectors to the extent that they allow for the detection and localization of objects in images. Furthermore, the proposed solution is extensible, as not only image modalities but also other sensor modalities can be considered according to the domain knowledge. We present the overall system and each of its components, and we stress the fusion approach.

### 3.1. Proposed Fusion Architecture

Domain knowledge in rail defect detection shows that using a single model to detect defects on rails is not sufficient. Firstly, to train such detectors, specific domain databases of very large scale have to be recorded and annotated; secondly, defect detection systems can make mistakes due to the physical nature of the defects. Thus, a simple spot on the rail can be confused with a rail break. Nevertheless, if such a rail imperfection appears together with some indicative context elements, such as missing nuts, for instance, it has to be detected as a defect. Thus, to solve the object (defect) classification problem, an architecture of two parallel branches is proposed. The first branch is designed for defect classification, while the second branch is used for classification of the clarifying context. The same model structure will be used for the first and second branches, but they are trained on different datasets. For the final decision, late fusion is applied since we need to provide a model-independent architecture. The proposed multibranch architecture is illustrated in Figure 3.

The first branch outputs an *object* class confidence vector $P(x_{i})$ containing probabilities of defect classes alone. Similarly, the second branch processes images to detect contextual elements and outputs a *context* class confidence vector $P(y_{j})$. Class confidence scores of these vectors are then fused as per the rules for the final decision-level fusion. The final defect and corresponding class confidence score $P(X_{d})$ are determined based on the decision rules. In this figure, the block “Decision-Making” implements late fusion in the decision space using the threshold-based rules. The probability threshold for elementary detection is denoted th in the figure. In the following, we detail the two branches of the architecture:**First branch: defect detection.** The first branch processes images for detecting the elements from the defect classes. The images are passed through the YOLOv5 with the CBAM module, resulting in feature vectors. These feature vectors are passed on to the classifier head for the final classification. The output is a class confidence vector $P(x_{i})$, where $0\leq P(x_{i})\leq 1$ and i=1,⋯,N, with *N* being the number of defect classes. Note that in our problem, N=4. The bounding boxes of the defects are also given by the YOLO detector.**Second branch: context detection.** In the second branch, the focus is on detecting contextual elements in the images. This branch is identical to the first branch, but the number of classes is different. The output of this branch is also a class confidence vector $P(y_{j})$, where $0\leq P(y_{j})\leq 1$ and j=1,⋯,M, with *M* being the number of classes of clarifying context elements. In our problem, M=6.

### 3.2. You Only Look Once Version 5 (YOLOv5)

The backbone of our solution is the YOLOv5 object detection network [44]. It is an impressive leap forward from its predecessors YOLOv3 and YOLOv4. This version excels in enhancing the speed and accuracy of tasks such as detection, classification, and segmentation. A noteworthy enhancement in YOLOv5 lies in its innovative use of adaptive image scaling and an adaptive anchor box, strategically optimizing the network’s efficiency. The scaling factor, derived from the ratio of the current image size (W and H), ensures a scaled size that minimizes computational load and allows detecting objects of different sizes, resulting in a substantial reduction in overall network calculation and improved performance. The YOLOv5 architecture is illustrated in Figure 4.

The YOLOv5 family comprises four models—*s*, *m*, *l*, and *x*—differing in width and depth. Among them, YOLOv5s is the lightest, featuring input, backbone (utilizing CSPDarknet53 [46]), neck (employing feature pyramid network and path aggregation network structures), and head (featuring a YOLOv3 anchor-based detection head). The CSPDarknet53 backbone extracts rich features, utilizing modules like focus and spatial pyramid pooling (SPP) to expedite training.

The backbone network is illustrated in Figure 4 in the upper dashed box. It consists of three types of blocks. The CBS block detailed in the middle on the right comprises convolutional layers, a batch normalization layer, and a nonlinearity SiLU. The latter is expressed as silu(x)=x∗σ(x), where σ(x) is the logistic sigmoid. The SPPF block comprises max-pooling operations and concatenating features after different steps in the max-pooling cascade, which allows for combining features of different resolutions. The C3 block is a simplified variant of cross stage partial (CSP) [47] blocks, which was created to increase the diversity of backpropagated gradients. The C3 block detailed in the lower left of Figure 4 comprises CBS blocks and bottlenecks. The bottleneck detailed in the lower right is a residual block, which has proven to be efficient since ResNet [48] architectures. The bottlenecks are just a sequence of two convolutions.

The neck block serves for building a pyramid of features. It is detailed in the center. The neck block consists of a cascade: C3, upsamplings, CBS, and concatenations.

The YOLOv5 head, responsible for detection, produces three sizes of feature maps—large (20 × 20), medium (40 × 40), and small (80 × 80). Here, the numbers express the coarseness of the detection grid, from the roughest to finest.

Utilizing three loss functions for classification, confidence, and location, YOLOv5 significantly enhances prediction accuracy through nonmaximum suppression (NMS). This synthesis highlights the YOLOv5s network structure, emphasizing its versatility with models ranging from nano- (YOLOv5n) to extra-large (YOLOv5x) size, each with varying trainable parameters. YOLOv5m, a medium-sized model with 21.2 million trainable parameters, is chosen in this study based on hardware constraints and database size considerations (approximately 20 k images for training and validation).

Although YOLOv5 performs well, reinforcement mechanisms, so-called attention modules, are interesting to add. We chose the convolutional block attention module (CBAM) [45] due to its ability to simultaneously amplify both the channels and the local features of those channels. The structure of the CBAM is presented in the following.

### 3.3. Convolutional Block Attention Module (CBAM)

Integration of attention blocks into deep NNs generally improves their performances, as channel attention and spatial attention allow for the selection of most significant feature channels and features both at the training and generalization steps [49]. The variety of attention modules, e.g., [50,51], usually comprise channel attention mechanisms, which weight or select the most important channels globally, and spatial attention mechanisms. The latter ensure the selection of important features in the channels with regard to the target objective function. Amongst them, we chose the CBAM, which comprises both mechanisms and is easy to integrate in the convolutional layers of convolutional neural networks (CNNs).

The CBAM [45] consists of two consecutive submodules: channel and spatial attention modules, see illustration in Figure 5.

For a given intermediate feature map, $F\in R^{c\times H\times W}$ with W×H spatial dimension and *c* number of channels as input, CBAM successively computes a 1D channel attention map $M_{c}\in R^{C\times 1\times 1}$. For each channel, a 2D spatial attention map, $M_{s}\in R^{1\times H\times W}$, is then computed as depicted in Figure 5. The overall attention process can be summarized with Equations (Equation 1) and (Equation 2) below:(1)$$F^{\prime }=M_{c}\left(F\right)⊗F$$
(2)$$F^{\prime \prime }=M_{s}\left(F^{\prime }\right)⊗F^{\prime }$$

Here, ⊗ denotes element-wise multiplication. During the multiplication process, attention values are appropriately broadcasted: channel attention values are distributed across the spatial dimension and vice versa. The ultimate refined output is represented by $F^{\prime \prime }$. Figure 6 illustrates the calculation procedure for each attention map. The subsequent sections provide an in-depth explanation of each attention module. As illustrated in Figure 6, the channel submodule makes use of both max-pooling and average-pooling outputs, which are processed by a shared network. Similarly, the spatial submodule takes advantage of the same two outputs, but these are pooled along the channel axis and then passed to a convolution layer [45].

**The channel attention map** is produced by exploiting the interchannel relationship of features. Each channel of a feature map is treated as a feature detector [52]; focusing on “**what**” is meaningful in the given input image. For efficient computation of channel attention, the spatial dimension of the input feature map is squeezed. To aggregate spatial information, average pooling has been commonly adopted so far. Zhou et al. [53] recommend it to learn the extent of the target object effectively, and it was adopted by Hu et al. [50] in their attention module to compute spatial statistics. In consideration of prior research, it has been argued that max pooling gathers another important clue about distinctive object features to infer finer channel-wise attention. Therefore, both average-pooled and max-pooled features are utilized simultaneously in the channel attention module, as empirically confirmed in [45]. This approach significantly improves the representation power of the neural network, demonstrating the effectiveness of the design choice. Firstly, the spatial information of a feature map is aggregated using both average-pooling and max-pooling operations, which generate two different spatial context descriptors: $F_{avg}^{c}$ for average-pooled features and $F_{avg}^{c}$ for max-pooled features, respectively. These descriptors are then forwarded to the shared multilayer perceptron (MLP) network with one hidden layer to generate the channel attention map $M_{c}\in R^{C\times 1\times 1}$. The hidden activation size is set to C/r×1×1 to reduce the parameter overhead, where *r* is the reduction ratio. After applying the shared network to each descriptor, the output feature vectors are merged using element-wise summation. By formula, the process of computing channel attention is summarized in Equation (Equation 3):
(3)$$\begin{array}{cc} M_{c}\left(F\right) & =\sigma (MLP(AvgPool(F))+MLP(MaxPool(F)))\\ \; & =\sigma (W_{1}\left(W_{0}\left(F_{avg}^{c}\right)\right)+W_{1}\left(W_{0}\left(F_{max}^{c}\right)\right)) \end{array}$$
where σ denotes the sigmoid function, $W_{0}\in R^{C/r\times C}$, and $W_{1}\in R^{C\times C/r}$. The MLP weights $W_{0}$ and $W_{1}$ are shared for both inputs, and the ReLU activation function is followed by $W_{0}$.**The spatial attention map** is generated by utilizing the interspatial relationship of features. The spatial attention focuses on “**where**” an informative part is, which is complementary to the channel attention. To compute the spatial attention, average-pooling and max-pooling operations are performed along the channel axis, and the outputs are concatenated to generate an efficient feature descriptor. Pooling operations along the channel axis is proven effective in highlighting informative regions [54]. On the concatenated feature descriptor, a convolution layer is applied to generate a spatial attention map $M_{s}\left(F\right)\in R^{H\times W}$ that encodes the areas to emphasize or suppress. The detailed operation is described below.The channel information of a feature map is aggregated by using two pooling operations, generating two 2D maps: $F_{avg}^{s}\in R^{1\times H\times W}$ for average-pooled features and $F_{max}^{s}\in R^{1\times H\times W}$ for max-pooled features across the channels. These features are then concatenated and convolved with a standard convolution layer, producing a 2D spatial attention map. By formula, the process of computing spatial attention can be represented as
(4)$$\begin{array}{cc} M_{s}\left(F\right) & =\sigma (f^{m\times m}\left(\left[AvgPool\left(F\right);MaxPool\left(F\right)\right]\right))\\ \; & =\sigma (f^{m\times m}\left(\left[F_{avg}^{s};F_{max}^{s}\right]\right)) \end{array}$$
where σ denotes the sigmoid function and $f^{m\times m}$ represents a convolution operation with a filter size of m×m. We keep the filter size of the original paper [45], m=7.

### 3.4. Modifications of Object Detector YOLOv5

In order to focus the YOLOv5m network to learn important features, we made two different types of network modifications in the baseline model. In the first modification, the CBAM was inserted at the ninth stage of the network, i.e., between the backbone and the neck, and in the second modification, the CBAM was plugged in between the neck and the detection head for the three different scales. Each modification has its own characteristics, as follows:**Type-I modification: CBAM at ninth stage.** The modification in the network architecture was performed on the baseline model mentioned above. A part of the C3-SPPF module was replaced with the CBAM, which reduced the computational overhead compared to the C3-SPPF module. Furthermore, it allowed the network to capture spatial dependencies and contextual information, allowing the model to have a better understanding of the image features.**Type-II modification: CBAM at three scales.** The SPPF module captures multiscale information from different levels of feature maps, which enhances the model’s ability to detect objects at various scales and sizes within an image. Thus, in the final architecture, we preserved the SPPF module at the ninth stage. As the CBAM proved to bring improvement in a monoscale setting, we introduced it at all three scales, i.e., small, medium, and large. The CBAM was added between the neck and the detection head for the respective scales, as shown in Figure 7. According to our experience, indeed, this network outperforms the previous ones.

The results obtained using this modified network (Type II) in comparison with the baseline YOLOv5m and Type-I modification are reported in the Results Section 5.1.

### 3.5. Decision Model: Rule-Based Decision-Level Fusion

The decision fusion model aims to combine the final decision scores of multiple individual models as per the rules defined in Table 1 to make a final, more accurate decision. Considering the problem of detecting rail defects, the goal of decision-level fusion is to leverage the unique strengths and capabilities of the individual object/context detection models to ultimately improve the overall performance and reliability of the decision-making process. Various fusion strategies are employed in decision fusion models, including equal-weighted voting, majority voting, weighted voting, etc. In our case, equal-weighted voting is considered, i.e., the decision scores from both the object and context branches are equally weighted. Table 1 outlines the decision rules from the domain knowledge for score fusion, considering the co-occurrence of various defect and context elements. In our task, four types of defects have to be detected, *defective fasteners*, *missing nuts*, *fishplate failure*, *and surface defects*, but only two defects need the context according to the domain-specific rules as depicted in Table 1. The defect detected based on the rules of the decision-level fusion model is considered the final defect, and its corresponding confidence score is denoted by $P(X_{d})$, where $0\leq P(X_{d})\leq 1$ and (1≤d≤4) as there are five types of final defects. The class confidence score $P(X_{d})$ of the final classified defect is equal to the confidence score $P(x_{i})$. Some defects require context under the rules in Table 1. Context’s score is $P(y_{j})$. $P(x_{i})$ and $P(y_{j})$ are above the threshold th. Here, the threshold th lies in the range 0≤th≤1. The four different types of defects classified by the decision model are mentioned below:*Defective fasteners*. The detection of defective fasteners in the image does not need image context. The final score is $P\left(X_{1}\right)=P\left(x_{1}\right)$.*Missing nuts*. To classify a defect as a missing nut, the defect detection branch of the proposed architecture in Section 3 should detect a missing nut in a fishplate with a confidence score $P(x_{2})\geq th$ AND the contextual (second) branch in Figure 3 should detect a fishplate with a confidence score $P(y_{2})\geq th$ AND the CES (clamp) context element should NOT be present or the confidence score for CES should be $P(y_{1})<th$. The corresponding class confidence score for missing nuts is $P\left(X_{2}\right)=P\left(x_{2}\right)$.*Surface defect*. The detection of a surface defect in the image does not need image context. The probability is $P\left(X_{3}\right)=P\left(x_{3}\right)$.*Fishplate failure*. To classify a defect as fishplate failure, a fishplate context element should be detected in the image with a confidence score of $P(y_{4})\geq th$. In this case, the class confidence score for fishplate failure $P(X_{4})$ is $P(y_{4})$.

## 4. Experiments

This section presents the different experiments performed on vision-based defect detection methods, i.e., training the baseline, Type-I, and Type-II models. Finally, it presents the decision-level fusion of the first and second branches of the fusion architecture proposed in Section 3.1.

### 4.1. Image Datasets

The images provided are used for training and validation of the baseline and the modified YOLO network. The images provided were RGB and grayscale images captured at 30 fps using a digital camera mounted on the front wheel of the data-recording train. The image data were collected over various numbers of train runs performed specifically for this project, and the whole image dataset can be broadly classified into two classes, the defect and context classes. These collected images contain various defects and context elements present on the rail lines. There is no preprocessing of images. It is not needed as the images are taken from a close camera view and the system has to adapt to different conditions, mainly lighting. Data augmentation is also not necessary because the images were taken under the same conditions, and the original image diversity is sufficient for good model training. As for image resolution, it ranges from 774×1480 up to 1508×1500, as the images were captured by HD cameras. In order to render the images of the normalized size for the YOLOv5 input, we subsampled them by interpolation to the size of 352×640 for 774×1480 and of 640×640 for 1508×1500. The resolution reduction ratios are thus quite reasonable, ranging from 1.88 to 2.35. Such a reduction preserves the objects of interest to detect, i.e., defects and context elements. Furthermore, low-pass filtering when resizing allows for reduction in impulse noise if it is present in the image. Details of both classes with the occurrence of samples are provided below:
**The defect set** has images that contain some defects present on the rail lines. The defects are *defective fasteners, surface defects,* or *missing nuts*. We note that, according to the domain knowledge, *fishplate failure* is at the same time a context element. It is annotated when the fishplate has a defect. The total number of provided examples and the number of examples used for training from each class is presented in Table 2. From the defect set, 9173 images were used for training the detector network, and 1368 images were used for validation. Note that in an image, we can find several defects. Defects are as follows:-*Defective fasteners*. These defects can pose significant safety risks and operational challenges. Fasteners are used to secure rail lines. They play a critical role in maintaining the integrity and stability of the railway infrastructure. Defective fasteners can lead to various issues, including track misalignment, track vibration, reduced load-bearing capacity, and safety hazards. The white frame in the center of Figure 8a shows an example of a defective fastener.-*Missing nuts*. The presence of missing nuts on rail lines refers to the absence of nuts that are used to secure bolts or fasteners in place along railway tracks, as shown by the two white frames in Figure 8c. This issue can occur due to various factors, such as inadequate maintenance, vibration, or mechanical failures.-*Surface defects*. These refer to irregularities or damage on the surface of the rails. These defects can occur due to various factors and pose significant risks to the safe operation of trains. Figure 8b shows an example of surface defects in the two white frames.-*Fishplate failure*. The error only manifests itself in the presence of context. It is in Figure 9a in the white frame.**The context set** contains images of the contextual elements, i.e., other essential elements present on the rail lines. The presence of these elements is considered while classifying a defect on the rail lines. Images in the context set can be classified into six categories: braid, CES (clamps), fishplate, seal (joints), welding, and markings. The total number of provided examples and the number of examples used for training from each class are presented in Table 3. In total, for the context set, 10,896 images were used for training the detector network, and 2940 images were used for validation. Note that in an image, we can find several contexts, including the following:-*Braid*. In the context of rail lines, a braid refers to the interwoven or interlaced strands of wire present between two joints. They can be seen as a series of parallel lines or ridges running along the length of the rail. Braids that are frayed or missing several strands are not considered defective. Figure 9d shows a braid in the white frame.-*CES (clamp)*. It is a contextual element present on the rail lines. Typically, the clamp is used at the joints of two rails in order to hold them together. An identified tightening CES in an image should invalidate the defects relating to missing nuts. Figure 9e shows a CES in the white frame.-*Fishplate*. Also known as a splice bar or joint bar, it is a metal plate used to connect two rails together at their ends, as shown in Figure 9a in the white frame. It is a critical component in maintaining the continuity and strength of the rail track. Fishplates provide stability, ensure proper alignment, and maintain the integrity of the track structure. Fishplates have a specific shape that conforms to the rail’s cross-sectional profile. They are usually rectangular or L-shaped with holes for bolts or other fastening mechanisms. The design may vary depending on the rail section and the type of fishplate being used. The fast movement of trains can cause loose or damaged bolts, wear on the plate surface, or cracks.-*Seal (joint)*. It is characterized by the presence of a discontinuity in the running surface and by the presence of a splice bar and/or braid. Seals are critical components in rail lines that allow for the connection of individual rail sections to create a continuous track. In traditional rail joints, a joint bar or fishplate is used along with bolts and nuts to connect adjacent rail sections. The joint bar is a metal plate that spans the rail ends, providing stability and maintaining alignment. Bolts and nuts secure the joint bar in place, ensuring a strong and reliable connection. The presence of seals along with a splice bar and/or a braid is sufficient to validate that it is not a rail-breaking defect. Figure 9a shows a seal in the red frame.-*Welding*. It joins rail sections together using heat and pressure to create a continuous and seamless track, as shown in Figure 9b. It eliminates the need for fishplates or other mechanical connectors. A weld is characterized by the presence of two parallel zones on the rail and a rib. The detection of welding in an image confirms the absence of a surface defect.-*Markings*. A defect already detected by an agent or a monitoring machine is marked using a painted symbol. This follow-up is materialized by the presence of a paint mark on the rail or sleeper, as shown in Figure 9c in the white frame. These marks can take various forms.

### 4.2. Experiments for Designing Vision-Based Defect and Context Detection

This section presents different experiment setups for vision-based defect detection methods, detailing the parameters used during the training. These experiments are performed in three parts: baseline, which is YOLOv5 without adding the CBAM block; CBAM at the 9th stage, which is the Type-I modification, see Section 3.4; and CBAM at three different scales, which is the Type-II modification as presented in Section 3.4. Since the defect and the context datasets had different numbers of classes, all three experiments were performed separately for the defect and context classifications. All calculations were performed using Python-3.10.13, torch-2.1.2, and an NVIDIA Tesla P100-PCIE-16GB. Parameters used during training for all architectures were as follows: batch size = 16, no. of epochs = 100, learning rate = 1 × $10^{-2}$. We limited the number of epochs to 100 as we observed the stabilization of loss function at this level. The batch size was also limited for the memory constraints, and the fixed learning rate was determined by a simple grid search. Training occurred as follows:*Baseline training*. In order to establish a baseline, the YOLOv5m [44] model, which was pretrained on the MS COCO [55] dataset that had 80 classes, was trained separately for defect and context elements.*Type-I model training*. In this experiment, the C3-SPPF module present at the 9th stage in the original YOLOv5m architecture was replaced with the CBAM, and this modified architecture was also trained separately for defect and context classes.*Type-II model training*. The CBAM was plugged in between the neck and the detection head for all three different scales: large (20 × 20), medium (40 × 40), and small (80 × 80). The modified architecture was trained separately for defect and context classes.

The corresponding results of this experiment with YOLOv5m are documented in Table 4 and Table 5 for defect and context classes, respectively. For comparison, the same tables contain the results of applying the above described architectures to YOLOv8m [32].

### 4.3. Decision-Level Fusion Threshold

Let us denote the defects that were correctly detected as true positive (*TP*). Defects that were not detected but are present in the images are false negative (*FN*). We compute the true-positive rate (TPR or recall (R)) as TP/(TP+FN). Defects that do not exist in the image but were detected by mistake are marked as false positive (*FP*). If there are no defects in the image and no defects were detected, such situations are marked as true negative (*TN*). Then the false-positive rate (FPR) is FP/(FP+TN). The precision (P) is calculated as TP/(TP+FP). Finally, accuracy (Acc) is calculated accordingly as (TP+TN)/(TP+TN+FP+FN). In Section 5, we show the obtained results of applying the threshold for Type II with the best results. In the same way, the ablation study will be demonstrated by removing the CBAM from the Type-II architecture.

### 4.4. Robustness Analysis Experiment

The manipulation of luminance levels is employed to emulate diverse ambient lighting conditions. To capture the subtle changes perceivable by the human visual system within midrange luminance (around 128), adjustments are made in increments of 10 levels. This approach spans a range from −30 to +30, corresponding to luminance reductions and increases, respectively, in steps of 10. The proposed methodology involves subjecting a test dataset to these controlled luminance modifications, specifically −30, −20, −10, 0 (original), 10, 20, and 30. This encompasses a range of variations up to approximately 10% of the luminance. Subsequently, our architectures are evaluated on this degraded test dataset to assess their robustness under varying lighting conditions. In Figure 10, we find a visual change in luminance.

The corresponding results of this experiment are documented in Section 5.3.

## 5. Results and Analysis

This section presents the various results obtained during the evaluation of the vision-based defect detection algorithms. First, the results of the architecture changes are demonstrated by adding a CBAM. After that, we evaluate the influence of the threshold th for all elementary detectors on the performance of the best architecture, Type II. Furthermore, we evaluate the improvement brought by the use of domain-knowledge fusion rules. After that, an ablation study is reported. Finally, the comparison with the SOTA and the complexity analysis are shown. A total of 4244 images were used in this experiment. The number of images with both context and defect is 66. The number of images with only context is 2875. The number of images with only a defect is equal to 1303. An example of the use of the YOLOv5 model can be found in Figure 11. Figure 11a contains missing nut defect detection, while Figure 11b presents the detected context elements. The probability scores are indicated in the figure. The used probability threshold for all elementary detectors is th=0.5.

In order to compare the performances of the *baseline* model YOLOv5 with the latest versions of this object detector, we conducted experiments with YOLOv8. As can be seen from Table 4, YOLOv8 does not strongly outperform YOLOv5 on our dataset, which is typical for the target application of railway inspection. Hence, in the precision metric, YOLOv8 improves only on 0.001–0.007 of our classes. In recall, YOLOv8 is lower for all classes but one. We stress that in our target application, recall is the most important metric for evaluation of the system performance. The performance in terms of mAP50 and map50–95 exhibits the same behavior. Hence, we cannot say that YOLOv8 outperforms YOLOv5. Furthermore, the YOLOv8m model contains 25,902,640 parameters, while YOLOv5m contains 20,861,016 parameters. Thus, the number of parameters of YOLOv8m is 24% higher compared to YOLOv5. In view of implementation on the embedded platform, the proposed system of the railway inspection algorithms should be as light as possible. Hence, we remain with our solution based on YOLOv5m. The overall performance of our system with fusion of object and context detection is presented in Table 6. In this table, we present the overall results of our systems (baseline, CBAM Type I, and CBAM Type II) on the dataset in the classification problem of three classes of defects: defective fastener, surface defect, and missing nut.

### 5.1. Results of Elementary Detectors with Added CBAM Blocks

This subsection compares the results in terms of precision (P); recall (R); mAP50* = mAP50 (±0.05), which is the mean average precision with 50% or higher IoU with the ground truth; and mAP50-95* = mAP50-95 (±0.05), which is a maP across the range of IoU from 50% to 95% [9]. The result is given as an ablation study, below:**Baseline vs. Type-I model**-*Detection of defect elements*. In Table 4a,b, we can compare the results obtained for the baseline model and Type-I model for the defect class for YOLOv5. It can be observed that the overall precision for the Type-I model is slightly reduced by ∼1.2% when compared to the baseline model. The reason for this reduction is that a part of the feature pyramid (C3-SPPF) module was replaced with the CBAM, resulting in a lesser number of features at different scales. However, there is an increase in the overall recall value of ∼0.8%, which implies that the Type-I model has a lower number of false negatives as the model learns and focuses on the most significant features. We state that for our problem of defect detection, the higher recall is better for the requirements of the whole industrial system. We can also observe in YOLOv8 a reduction in overall precision (∼0.1%) and an increase in the overall recall (∼0.2%).-*Detection of context elements*. Similarly, for context elements, in Table 5a,b, we can see the results obtained for the baseline model and Type-I model for YOLOv5. It can be observed that even though the precision for the Type-I model is close to the baseline model, there is an increase of ∼0.7% in the overall recall of the Type-I model, indicating that it generates fewer false negatives than the baseline model. In general, the overall precision of the Type-I model has slightly gone down because part of the feature pyramid (C3-SPPF) module was replaced with the CBAM, which decreases the number of features at different scales. In YOLOv8, the overall precision has an increase of ∼0.2%.**Type-I vs. Type-II model**-*Detection of defect elements*. With Table 4b,c, we can compare the metrics for the Type-I and Type-II models on defect class elements. It can be noticed that the overall performance for the Type-II model increases when compared to the Type-I model and the baseline model. For the Type-II model, the overall precision increases by ∼1.94% and the overall recall increases by ∼0.9% when compared to the Type-I model, and there is an increase of ∼ 0.6% in the overall precision and an increase of ∼1.7% in the overall recall value when compared to the baseline model. This improvement validates the idea that modifying the YOLOv5 network by including a CBAM at the detection stage for each scale (small, medium, and large) should enhance the overall performance of the model, as it helps the model to focus on and learn the most significant features. For YOLOv8, we can see that overall precision for Type II decreases by ∼0.9%, but overall recall is increased by ∼1.3% when we compare it to the Type-I model. If we compare it to the baseline model, we find for the overall precision a decrease of ∼1.0% and for the overall recall an increase of ∼1.6%.-*Detection of context elements*. Similarly, Table 5b,c present the results for evaluation of context class elements and comparison of the Type-I and Type-II models’ results. It can be observed in Table 5c that the overall performance metrics of the Type-II model improve when compared to the Type-I model and the baseline model. The Type-II model has an increase of ∼1.12% in the overall precision value and an increase of ∼0.21% in the overall recall value when compared to the Type-I model. Furthermore, there is an increase of ∼0.9% in the overall precision and an increase of ∼0.96% in the overall recall value when compared to the baseline model. Therefore, these improved results of the Type-II model verify the objective of modifying the YOLOv5 network by including a CBAM at three detection scales, which should enhance the model’s overall performance. Further, in comparing the baseline model and the Type-II model, it can be noticed that the recall increases further, implying that the Type-II model has further reduced the Type-II error, i.e., reduced false negatives, as the model focuses mainly on the most significant features. For YOLOv8, we can see that overall precision for Type-II decreases by ∼0.5% and overall recall decreases by ∼0.4% when we compare it to the Type-I model. If we compare it to the baseline model, we find for the overall precision a decrease by ∼0.3% and for the overall recall a decrease by ∼0.7%.

### 5.2. Ablation Study

In order to check the performance of our solution, in this section, we conducted ablation studies for YOLOv5m.

*CBAM Type II with fusion*. Figure 12 illustrates the selection of different threshold values (from 0.0 to 1.0) for all elementary detectors of the Type-II CBAM architecture for defects and context elements. These results are obtained after the application of the fusion rules described in Section 3.5. For elementary detectors with fusion, using the threshold th value of 0.82, the resulting accuracy (Acc) is 0.7955. The precision (P) value is 0.7640. The recall (R) or true-positive rate (TPR) is equal to 0.6436. The false-positive rate (FPR) is equal to 0.1159.*CBAM Type II*. The results obtained without applying the rules are illustrated in Figure 13. After removing the fusion rules, the accuracy was decreased by 0.0026 and is 0.7929. This is due to the slight decrease in precision (by 0.0082), which is 0.7558, and to the rise in FPR (by 0.0056 and is 0.1215). Consequently, the recall or true-positive rate value increased by 0.0023 and is equal to 0.6459.*CBAM Type I with fusion*. Using the threshold th value of 0.78, the resulting accuracy (Acc) is 0.7948. The precision (P) value is 0.6935. The recall (R) or true-positive rate (TPR) is equal to 0.7773. The false-positive rate (FPR) is equal to 0.1953. Figure 14 illustrates these results.*CBAM Type I*. Without applying the rules, the accuracy decreased by 0.0059 and is 0.7889. The precision decreased by 0.0153 and is equal to 0.6782. The false-positive rate increased by 0.0139 and is equal to 0.2092. The recall or true-positive rate also increased by 0.0082 and is equal to 0.7855. We can find these results in Figure 15.*Baseline with fusion*. Using the threshold th value of 0.80, the resulting accuracy (Acc) is 0.8032. The precision (P) value is 0.7122. The recall (R) or true-positive rate (TPR) is equal to 0.7698. The false-positive rate (FPR) is equal to 0.1777. Figure 16 illustrates these results.*Baseline*. Without applying the rules, the accuracy decreased by 0.0057 and is 0.7975. The precision decreased by 0.0153 and is equal to 0.6969. The false-positive rate increased by 0.0137 and is equal to 0.1914. The recall or true-positive rate also increased by 0.0081 and is equal to 0.7779. We can find these results in Figure 17.

In the case of removing the CBAM elements from the architecture, applying the threshold to all elementary detectors leads to a slightly better accuracy result. Indeed, with our best architecture, Type II, the accuracy is 0.7929, while without introduction of the CBAM, the accuracy is 0.7975. Moreover, the location of the CBAM also affects the result. Type I was the worst of all, with the accuracy 0.7889. This means that the YOLO detectors were sufficiently well tuned, and for the given validation threshold, the adding of attention modules to these networks might not be necessary. The results of the ablation study are presented in Table 6.

It can be seen that removing fusion from all three types of architectures, see Table 6, columns 3, 5, 7, decreases the accuracy. Therefore, we state that our hybrid system with domain-knowledge rules induces a noise-filtering effect in the decision space. A small drop in recall in Type II, Type I, and baseline with fusion is not critical in this case.

### 5.3. Robustness Analysis

This section presents the results of the study of our system’s resistance to degradation, namely, to changes in lighting conditions.

According to the results in Table 7, we can see small changes not exceeding 1% for accuracy; for P, it is 3%, and for R it is 6%. For FPR, we can find changing by 15% between +20 and +10 for baseline + fusion and baseline.

According to the results in Table 8, we can see small changes not exceeding 1% for accuracy. For P, it can be 1% for Type I + fusion, and for Type I, it is 12%. For R, it is 4% for Type I + fusion. For Type I, we have 28%, changing between −30 and +30. For FPR in Type I + fusion, we can find changing by 9%, but in Type I, between −30 and +30, it is 80%.

According to the results in Table 9, we can see small changes not exceeding 1% for accuracy. For P, it can be 10% for Type II + fusion and for Type II, it is 3%. For R, it is 21% for Type II + fusion between -20 and −10. For Type II, we have 5%. For FPR in Type II + fusion, we can find changing by 69% between +20 and 0 but in Type II, it is 19%.

Based on the above results, we confirm that such changes as in lighting conditions do not affect the performance of YOLOv5. Moreover, using fusion still improves the accuracy.

### 5.4. Comparison with the State of the Art

A direct comparison with the SOTA in rail defect detection is not possible, as we propose the first system based on the specific domain knowledge (rules) and database; due to the stringent IPR constraints, these cannot be shared for a comparative experiment. Nevertheless, it is important to position our contribution. Thus, we take as reference the paper by Li et al. [56], which is also devoted to the detection of defects on the rails. They introduced an ensemble learning model designed to enhance predictive performance through the integration of multiple learning algorithms. The authors employed distinct backbone neural networks individually to extract features, subsequently blending these features in a binary format to generate diverse and enhanced subnetworks.

When applied to an eight-class defect dataset, the proposed multibackbone double augmentation (MBDA) system demonstrated improvements in mean average precision at 0.5 intersection over union (mAP@.5). Specifically, it achieved a 0.045 higher mAP@.5 compared to the faster R-CNN model and 0.053 higher mAP@.5 when contrasted with YOLOv5 on the validation dataset. Note that we use the same version of YOLO. They give the performance estimates in terms of mAP@.5, which is a more permissive metric. Using our domain-knowledge fusion rules, we increased the accuracy by 0.0057. This is not a strong increase, but taking into account the requirements of a light system design and multimodal general framework, we still consider this result as appropriate.

### 5.5. Complexity Analysis

As the proposed defect detection system has to run on embedded hardware, we perform both complexity analyses for temporal and spatial parameters, that is, execution time and model volume. In Table 10, the computational times for the YOLOv5m detector are given for different system configurations, all run on an NVIDIA Tesla P100-PCIE-16G GPU. For spatial complexity, which is the model volume, we give the estimations in terms of number of parameters and model size. The training of the baseline model on the defect dataset took 8.1 h, and on the context it took 10 h for 100 epochs. The training time for the Type-I model for the defect dataset took 7.8 h, and the context class took 9.7 h for 100 epochs. The Type-II model took 10 h of training time for the defect class and 12.4 h for the context class, each for 100 epochs. From the first and second columns of the table, one can see that the computational times at inference for object and context are the same as for the images are of a normalized dimension. Generally, the times range from 0.0124 to 0.0158 as a function of the system complexity. The lightest configuration is the one of CBAM Type I where we replace the ninth block of YOLOv5m with the CBAM. The heaviest computational times occur when the CBAM is inserted as a complementary module (Type II). This holds for the number of parameters and the volume of the models. Resulting from the performance analysis, see Section 5.2, we conclude that introduction of a CBAM does not strongly improve the model. Therefore, when the model volume is a critical metric for integration into an embedded system, then CBAM Type-I integration may be performed. When the performance is the absolute criterion, then the system should be used in its baseline or CBAM Type-II (for improved precision metric) configuration.

The fusion times are negligible compared to the YOLOv5m inference times and are of $9.775161743164062\times 10^{\left(-6\right)}$. As object detection and context detection might be run in parallel, then this time will be added to the inference time of object (defect) detection. These times are already compatible with real time.

## 6. Conclusions

In this work, we introduced an innovative hybrid approach to address the challenges associated with defect detection on rail lines. It is based on the elementary object and context detectors of the YOLO family, which uses a deep learning approach, and on domain-knowledge fusion rules in the decision space. We showed that the usage of domain-knowledge rules allows for removing limitations inherent to singular detectors, which often yield unreliable outcomes.

Our proposed architecture adopts a late-fusion technique, specifically a rule-based decision-level fusion. This involves the joint analysis of the confidence scores derived from multiple individual models through a predefined set of rules. This form of fusion affords the flexibility to incorporate various detection models or modify their quantities by introducing additional sensor modalities for defect detection in accordance with domain-specific knowledge rules.

Firstly, we tried to improve YOLO detectors by incorporating a powerful CBAM attention model in two different places in the YOLO: replacement of the C3-SPPF (Type I) and a multiscale inclusion before the decision heads (Type II). Indeed, the Type-II architecture brought a slight improvement in both recall and precision metrics, circa 1%. Nevertheless, when searching for the optimal classification score threshold for the target evaluation after fusion, we stated that the incorporation of a CBAM model was not necessary for well-tuned YOLO detectors.

Next, we stated that fusion on the basis of domain-knowledge rules has a noise-filtering effect in the decision space and allows for a slight increase in the precision of the overall system.

Last but not least, we thus built our hybrid system, which allows for an easy plugging in of different modalities and deep learning frameworks merely as black boxes, as the fusion for reliable decisions occurs in the decision space. This facilitates the use of our system in real-world industrial applications.

## Figures and Tables

**Figure 1 sensors-24-01171-f001:**
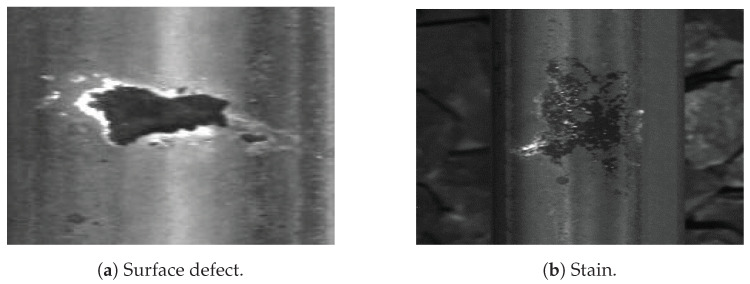
Example of contextual dependencies.

**Figure 2 sensors-24-01171-f002:**
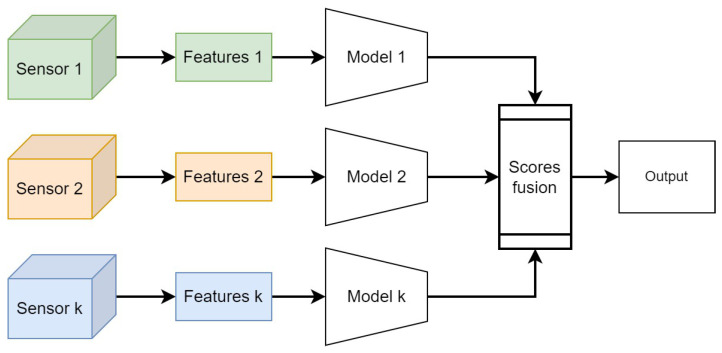
Late-fusion strategies integrate decisions made by submodels specific to each sensor.

**Figure 3 sensors-24-01171-f003:**
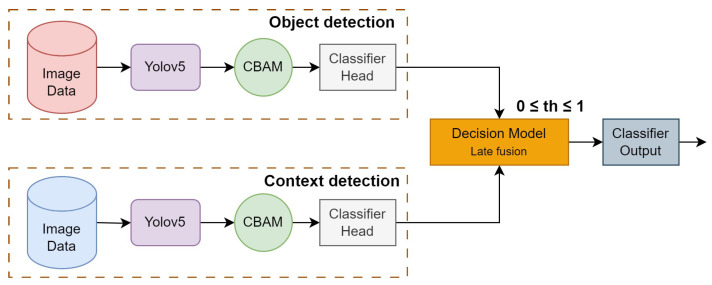
Layered architecture with decision rules for defect detection.

**Figure 4 sensors-24-01171-f004:**
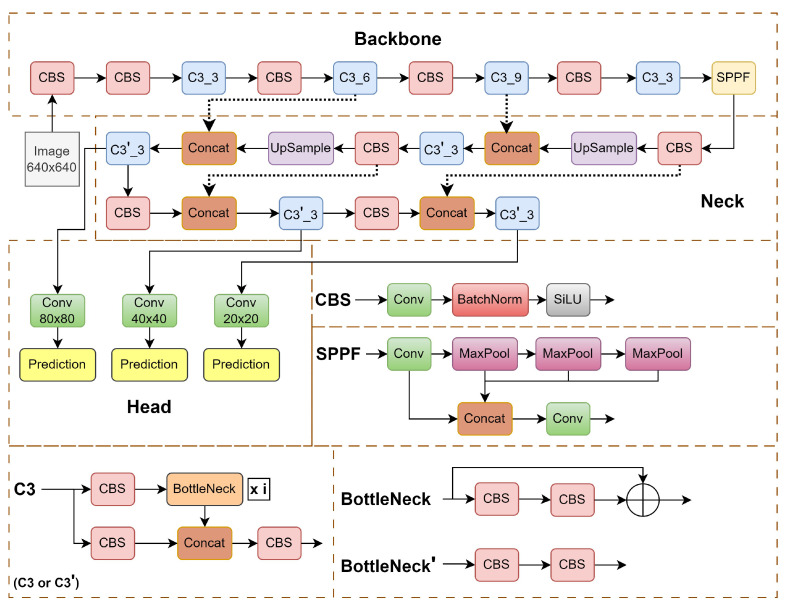
YOLOv5 network architecture. It mainly includes three parts: the backbone network, the neck, and the detection head [44].

**Figure 5 sensors-24-01171-f005:**
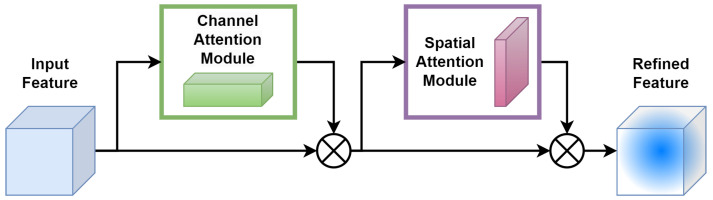
Architecture of convolutional block attention module (CBAM).

**Figure 6 sensors-24-01171-f006:**
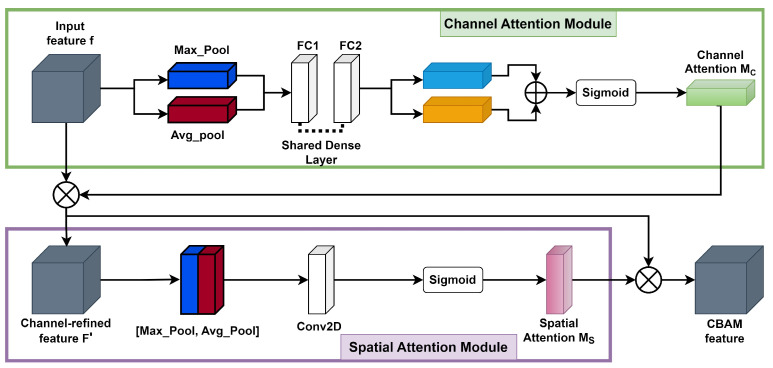
Representation of each attention submodule in CBAM.

**Figure 7 sensors-24-01171-f007:**
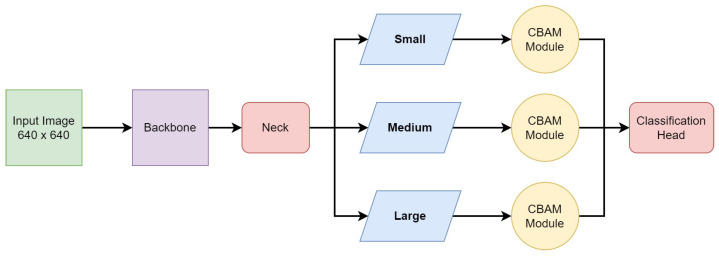
The modified YOLOv5 network architecture, where the CBAM is added between the neck and the detection head for the three respective scales, i.e., small, medium, and large scales.

**Figure 8 sensors-24-01171-f008:**
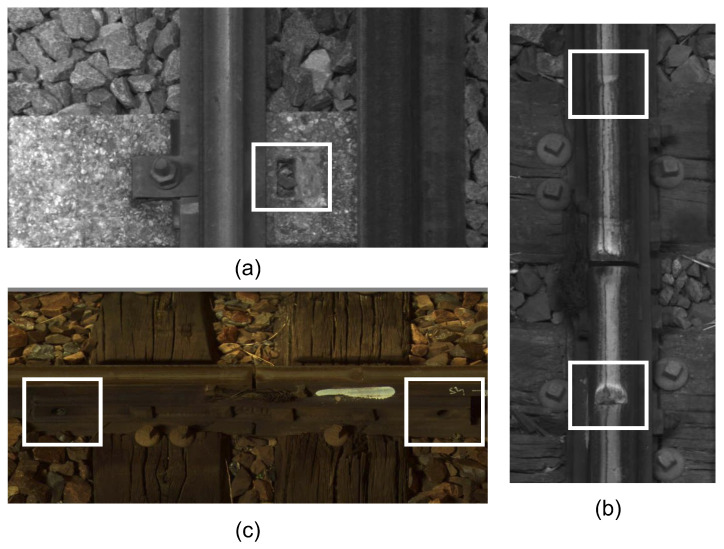
Examples of possible defects (highlighted with white boxes). (**a**) Defective fastener, (**b**) surface defects, (**c**) missing nuts.

**Figure 9 sensors-24-01171-f009:**
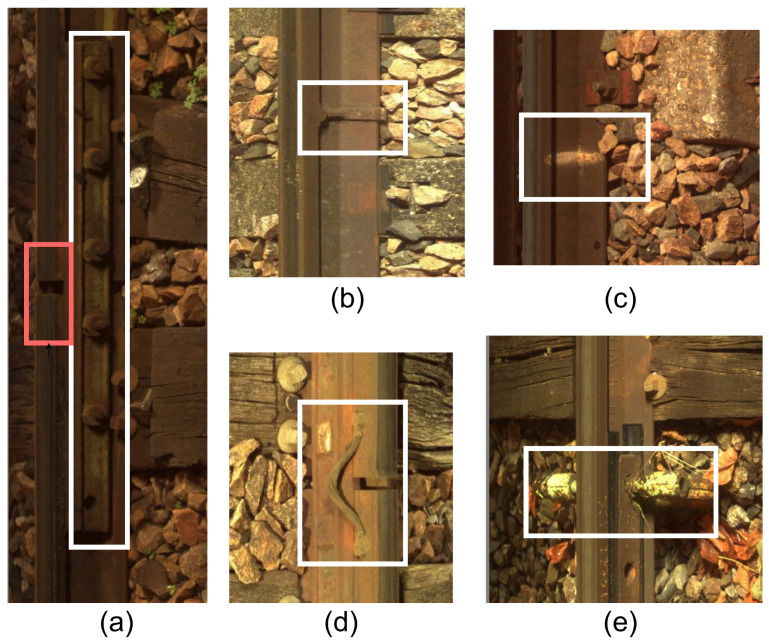
Images of possible contexts (highlighted with colored frames). (**a**) Fishplate (white frame) and seal (red frame), (**b**) welding, (**c**) markings, (**d**) braid, (**e**) CES.

**Figure 10 sensors-24-01171-f010:**
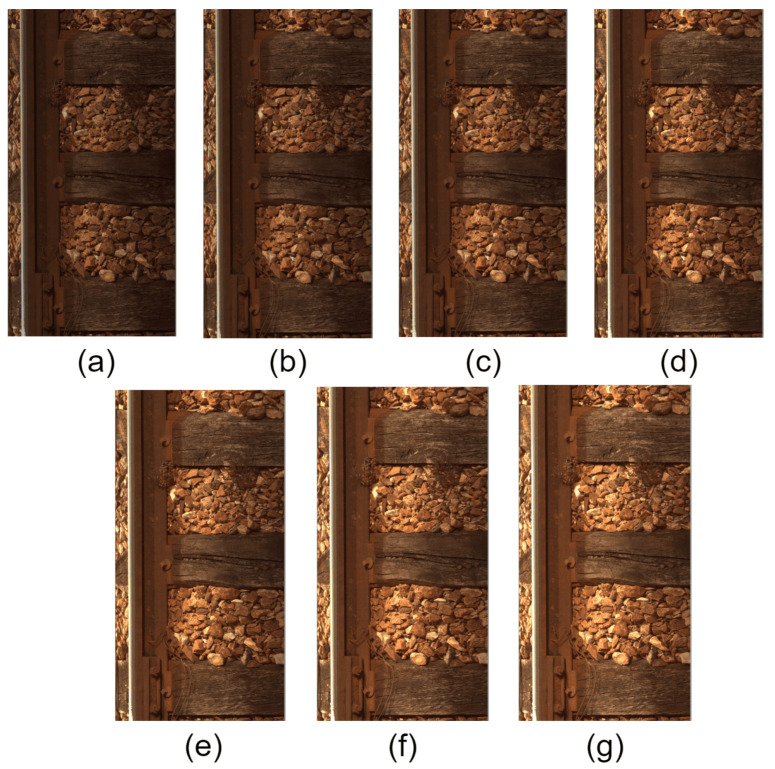
Image of changes in luminance: (**a**) −30, (**b**) −20, (**c**) −10, (**d**) 0, (**e**) +10, (**f**) +20, (**g**) +30.

**Figure 11 sensors-24-01171-f011:**
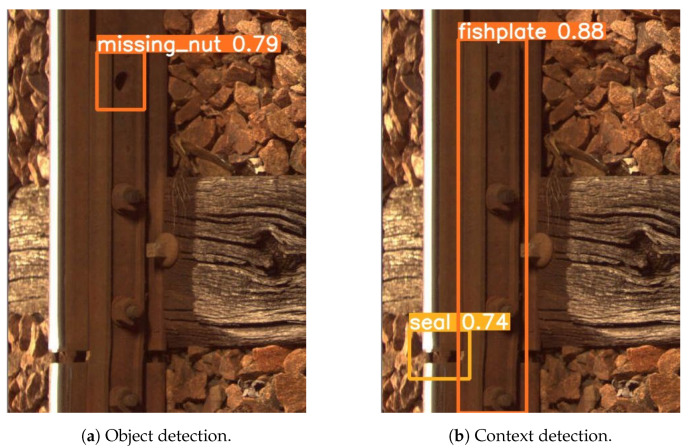
An example of object and context detection with YOLOv5.

**Figure 12 sensors-24-01171-f012:**
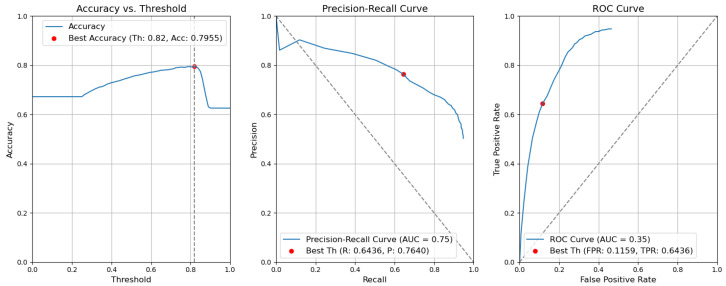
Accuracy, precision–recall, and ROC curves after applying rules using YOLOv5 Type-II models for object and context.

**Figure 13 sensors-24-01171-f013:**
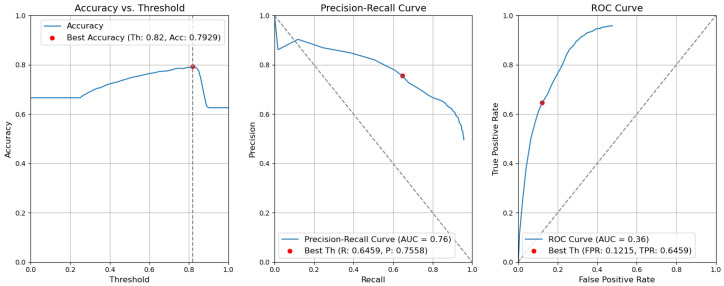
Accuracy, precision–recall, and ROC curves without applying rules using YOLOv5 Type-II models for object and context.

**Figure 14 sensors-24-01171-f014:**
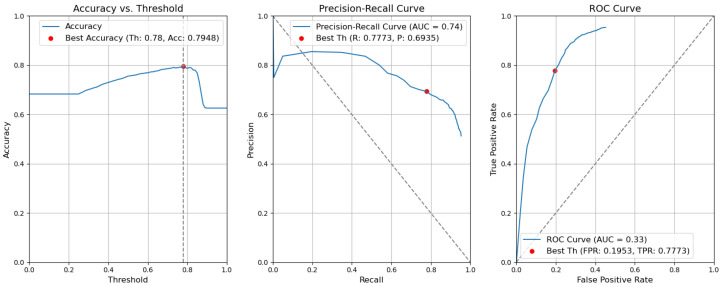
Accuracy, precision–recall, and ROC curves after applying rules using YOLOv5 Type-I models for object and context.

**Figure 15 sensors-24-01171-f015:**
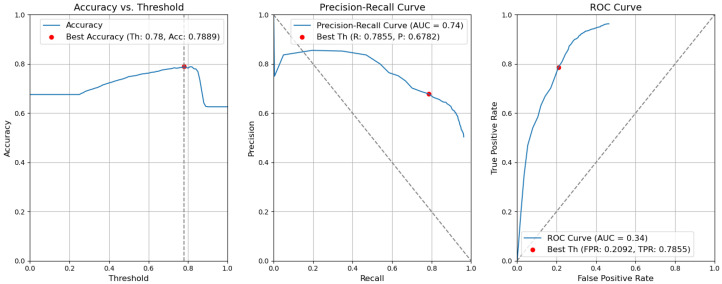
Accuracy, precision–recall, and ROC curves without applying rules using YOLOv5 Type-I models for object and context.

**Figure 16 sensors-24-01171-f016:**
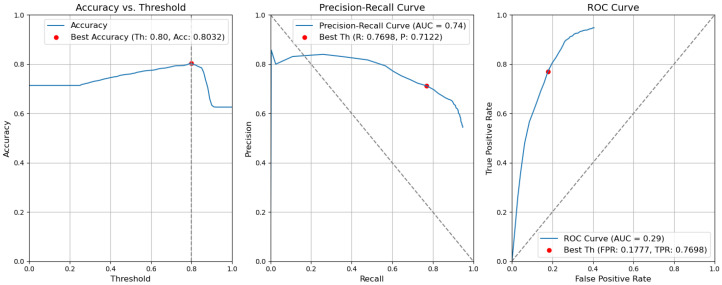
Accuracy, precision–recall, and ROC curves after applying rules using YOLOv5 baseline models for object and context.

**Figure 17 sensors-24-01171-f017:**
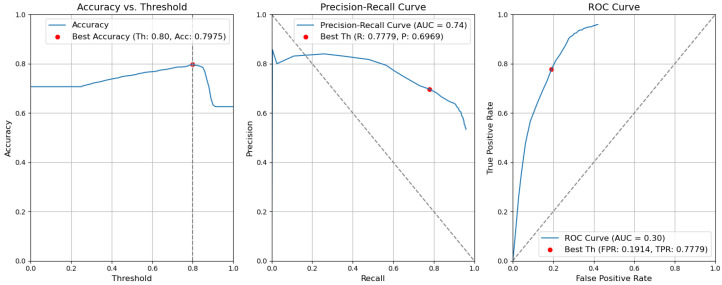
Accuracy, precision–recall, and ROC curves without applying rules using YOLOv5 baseline models for object and context.

**Table 1 sensors-24-01171-t001:** Rules to be considered by the decision model to classify the different types of defects that require context.

Defects Detected	Contextual Elements Detected	Decision Made by the System		
		Missing Nuts	Fishplate Failure	No Defect
Missing nuts	Fishplate (image)	X		
	CES (image)			X
Fishplate failure	Fishplate (image)		X	

**Table 2 sensors-24-01171-t002:** The total number of examples provided for training of the defect element detector.

Class	Used Examples
Defective_fastener	8127
Surface_defect	2679
Missing_nut	1490
Total	12,296

**Table 3 sensors-24-01171-t003:** The total number of examples provided for training of the context element detector.

Class	Used Examples
Braid	3783
CES	577
Fishplate	5923
Seal	6441
Welding	2837
Markings	2820
Total	23,391

**Table 4 sensors-24-01171-t004:** Comparing the precision (P), recall (R), mAP50* = mAP50 (±0.05), and mAP50-95* = mAP50-95 (±0.05) values between the baseline and Type-I models trained on defect images. Overall, this represents the metrics for all the classes combined.

**(a) YOLOv5m (Baseline) and YOLOv8m (Baseline)**								
**Model Type**	**YOLOv5 (Baseline)**				**YOLOv8 (Baseline)**			
**Class**	**P**	**R**	**mAP50***	**mAP50-95***	**P**	**R**	**mAP50***	**mAP50-95***
Overall	0.886	**0.880**	**0.901**	**0.468**	**0.887**	0.865	0.877	0.466
Defective fastener	0.934	**0.962**	**0.961**	0.632	**0.935**	0.960	0.961	**0.644**
Surface defect	0.814	**0.764**	**0.848**	**0.340**	**0.818**	0.720	0.793	0.326
Missing nut	0.902	**0.916**	**0.895**	**0.431**	**0.909**	0.916	0.875	0.427
**(b) YOLOv5m with CBAM at 9th Stage (Type I) and YOLOv8m with CBAM at 9th Stage (Type I)**								
**Model Type**	**YOLOv5 with CBAM at 9th Stage (Type I)**				**YOLOv8 with CBAM at 9th Stage (Type I)**			
**Class**	**P**	**R**	**mAP50***	**mAP50-95***	**P**	**R**	**mAP50***	**mAP50-95***
Overall	0.875	**0.887**	**0.895**	0.455	**0.886**	0.867	0.873	**0.464**
Defective fastener	0.936	**0.967**	0.963	**0.634**	**0.939**	0.963	**0.967**	**0.648**
Surface defect	0.789	**0.784**	**0.831**	0.310	**0.801**	0.718	0.757	**0.319**
Missing nut	0.900	0.910	0.892	0.421	**0.919**	**0.921**	**0.896**	**0.426**
**(c) YOLOv5m with CBAM at Three Scales (Type II) and YOLOv8m with CBAM at Three Scales (Type II)**								
**Model Type**	**YOLOv5 with CBAM at Three Scales (Type II)**				**YOLOv8 with CBAM at Three Scales (Type II)**			
**Class**	**P**	**R**	**mAP50***	**mAP50-95***	**P**	**R**	**mAP50***	**mAP50-95***
Overall	**0.892**	**0.895**	**0.903**	**0.469**	0.878	0.879	0.882	0.469
Defective fastener	**0.942**	**0.968**	0.965	0.633	0.926	0.964	**0.967**	**0.651**
Surface defect	**0.832**	**0.795**	**0.840**	**0.361**	0.795	0.752	0.800	0.330
Missing nut	0.910	**0.921**	**0.904**	0.413	**0.914**	0.921	0.880	**0.425**

**Table 5 sensors-24-01171-t005:** Comparing the precision (P), recall (R), mAP50* = mAP50 (± 0.05), and mAP50-95* = mAP50-95 (± 0.05) between the baseline and Type-I models trained on context images. Overall, this represents the metrics for all the classes combined.

**(a) YOLOv5m (Baseline) and YOLOv8m (Baseline)**								
**Model Type**	**YOLOv5 (Baseline)**				**YOLOv8 (Baseline)**			
**Class**	**P**	**R**	**mAP50***	**mAP50-95***	**P**	**R**	**mAP50***	**mAP50-95***
Overall	0.893	0.932	0.923	0.492	**0.902**	**0.942**	**0.926**	**0.493**
Braid	0.880	0.953	0.928	0.530	**0.898**	**0.964**	**0.963**	**0.574**
CES	0.918	**0.983**	**0.984**	**0.558**	**0.940**	0.974	0.976	0.558
Fishplate	**0.932**	0.936	**0.968**	**0.627**	0.920	**0.957**	0.953	0.559
Seal	0.859	**0.877**	0.848	0.329	**0.866**	0.876	**0.864**	**0.349**
Welding	0.906	0.943	**0.916**	0.441	**0.913**	**0.953**	0.912	**0.443**
Markings	0.860	**0.938**	0.886	0.470	**0.875**	0.927	**0.887**	**0.473**
**(b) YOLOv5m with CBAM at 9th Stage (Type I) and YOLOv8m with CBAM at 9th Stage (Type I)**								
**Model Type**	**YOLOv5 with CBAM at 9th Stage (Type I)**				**YOLOv8 with CBAM at 9th Stage (Type I)**			
**Class**	**P**	**R**	**mAP50***	**mAP50-95***	**P**	**R**	**mAP50***	**mAP50-95***
Overall	0.891	**0.939**	0.918	0.489	**0.904**	0.939	**0.932**	**0.517**
Braid	0.878	0.953	0.926	**0.528**	**0.892**	**0.958**	**0.957**	**0.574**
CES	0.920	**0.974**	**0.981**	0.554	**0.934**	0.969	0.975	**0.575**
Fishplate	0.930	0.940	0.962	0.624	**0.947**	**0.978**	**0.984**	**0.679**
Seal	0.861	**0.874**	0.841	0.325	**0.865**	0.860	**0.862**	**0.351**
Welding	0.901	**0.954**	0.912	0.440	**0.915**	0.951	**0.920**	**0.445**
Markings	0.856	**0.940**	0.889	0.466	**0.870**	0.918	**0.895**	**0.477**
**(c) YOLOv5m with CBAM at Three Scales (Type II) and YOLOv8m with CBAM at Three Scales (Type II)**								
**Model Type**	**YOLOv5 with CBAM at Three Scales (Type II)**				**YOLOv8 with CBAM at Three Scales (Type II)**			
**Class**	**P**	**R**	**mAP50***	**mAP50-95***	**P**	**R**	**mAP50***	**mAP50-95***
Overall	**0.901**	**0.941**	**0.924**	0.493	0.899	0.935	0.923	**0.495**
Braid	**0.900**	0.955	0.933	0.515	0.893	**0.962**	**0.954**	**0.578**
CES	0.924	**0.983**	**0.986**	0.561	**0.940**	0.974	0.969	**0.569**
Fishplate	**0.939**	0.941	**0.963**	**0.628**	0.916	**0.946**	0.949	0.563
Seal	**0.868**	**0.878**	0.848	0.330	0.862	0.862	**0.865**	**0.450**
Welding	**0.914**	**0.955**	**0.925**	**0.454**	0.908	0.946	0.906	0.439
Markings	0.861	**0.942**	**0.894**	0.469	**0.876**	0.922	0.892	**0.474**

**Table 6 sensors-24-01171-t006:** Ablation study results.

	Architecture					
**Metric**	**Type II + Fusion**	**Type II**	**Type I + Fusion**	**Type I**	**Baseline + Fusion**	**Baseline**
Acc	0.7955	0.7929	0.7948	0.7889	**0.8032**	0.7975
P	**0.7640**	0.7558	0.6935	0.6782	0.7122	0.6969
R or TPR	0.6436	0.6459	0.7773	**0.7855**	0.7698	0.7779
FPR	0.1159	0.1215	0.1953	**0.2092**	0.1777	0.1914

**Table 7 sensors-24-01171-t007:** Robustness analysis for baseline + fusion and baseline architectures.

	Baseline + Fusion				Baseline			
	**Acc**	**P**	**R**	**FPR**	**Acc**	**P**	**R**	**FPR**
−30	0.8024	0.7152	0.7593	0.1730	0.7967	0.6999	0.7663	0.1861
−20	0.8006	0.7115	0.7599	0.1761	0.7949	0.6964	0.7669	0.1892
−10	0.8013	0.7110	0.7640	0.1773	0.7961	0.6964	0.7721	0.1904
0	0.8032	0.7122	0.7698	0.1777	0.7975	0.6969	0.7779	0.1914
+10	0.8016	0.7089	0.7703	0.1806	0.7959	0.6940	0.7776	0.1940
+20	0.8010	0.7263	0.7297	0.1580	0.7965	0.7140	0.7343	0.1680
+30	0.8003	0.7241	0.7308	0.1598	0.7958	0.7119	0.7355	0.1698

**Table 8 sensors-24-01171-t008:** Robustness analysis for Type-I + fusion and Type-I architectures.

	Type I + Fusion				Type I			
	**Acc**	**P**	**R**	**FPR**	**Acc**	**P**	**R**	**FPR**
−30	0.7964	0.6873	0.8023	0.2070	0.7903	0.6723	0.8110	0.2213
−20	0.7936	0.6836	0.8000	0.2100	0.7878	0.6786	0.7797	0.2077
−10	0.7957	0.6965	0.7738	0.1919	0.7900	0.6812	0.7826	0.2059
0	0.7948	0.6935	0.7773	0.1953	0.7889	0.6782	0.7855	0.2092
+10	0.7949	0.6938	0.7772	0.1951	0.7888	0.6782	0.7855	0.2090
+20	0.7946	0.6937	0.7767	0.1952	0.7892	0.7298	0.6738	0.1442
+30	0.7932	0.6923	0.7733	0.1954	0.7877	0.7500	0.6331	0.1226

**Table 9 sensors-24-01171-t009:** Robustness analysis for Type-II + fusion and Type-II architectures.

	Type II + Fusion				Type II			
	**Acc**	**P**	**R**	**FPR**	**Acc**	**P**	**R**	**FPR**
−30	0.7954	0.7120	0.7372	0.1712	0.7924	0.7590	0.6390	0.1182
−20	0.7956	0.7107	0.7413	0.1732	0.7920	0.7562	0.6419	0.1206
−10	0.7950	0.7636	0.6424	0.1160	0.7925	0.7554	0.6448	0.1215
0	0.7955	0.7640	0.6436	0.1159	0.7929	0.7558	0.6459	0.1215
+10	0.7953	0.7627	0.6448	0.1169	0.7931	0.7556	0.6471	0.1219
+20	0.7951	0.6936	0.7791	0.1958	0.7933	0.7785	0.6151	0.1025
+30	0.7942	0.7608	0.6436	0.1180	0.7923	0.7757	0.6151	0.1040

**Table 10 sensors-24-01171-t010:** Computational complexity analysis of YOLOv5.

	Baseline (Object)	Baseline (Context)	CBAM Type I (Object)	CBAM Type I (Context)	CBAM Type II (Object)	CBAM Type II (Context)
Training time (h)	8.1	10	7.8	9.7	10	12.4
Inference speed (s): image resolution (774 × 1480)	0.0090	0.0090	0.0085	0.0085	0.0113	0.0113
Inference speed (s): image resolution (1508 × 1500)	0.0124	0.0124	0.0119	0.0119	0.0158	0.0158
No. of parameters	20,861,016	20,873,139	17,319,098	17,331,221	21,480,702	21,489,369
Model size (MB)	40.1	40.2	33.3	33.4	41.3	41.3

## Data Availability

Data are the exclusive property of SNCF Réseau and may not be distributed under any circumstances.

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
