# Peer review of "A Hybrid System for Defect Detection on Rail Lines through the Fusion of Object and Context Information"

_sensors, 2024, doi:10.3390/s24041171_

Round 1

Reviewer 1 Report

Comments and Suggestions for Authors

This article uses YOLOv5 to jointly analyze context to form multimodal fusion and defect prediction decisions. The technical route is reasonable and feasible. It is recommended to supplement the comparison of RCNN algorithms and YOLO8's latest achievement in algorithm comparison.

Comments on the Quality of English Language

Pay attention to the tense.

Author Response

It is recommended to supplement the comparison of RCNN algorithms and YOLO8's latest achievement in algorithm comparison.

Thank you for your valued comment. We conducted experiments using Yolov8. The results of this experiment are described in Section 5 in different tables. 4 and 5.

The authors of "A Comparison of YOLOv5 and YOLOv8 in the Context of Mobile UI Detection" state that Yolov8 does not always outperform YOLOv5. Furthermore, YOLOv8m contains 25902640 parameters to train compared to 20861016 parameters, which is more difficult to train. Therefore, in our work, we remain with YOLOv5 and present other results with this object detector.

Concerning Faster RCNN,  in "Aerial Images Processing for Car Detection using Convolutional Neural Networks: Comparison between Faster R-CNN and
YoloV3"  and in "Smart Pothole Detection Using Deep Learning Based on Dilated Convolution" the authors report, that this Faster R-CNN not always outperform even YOLO. Therefore, we remain with our solution based on YOLOv5 which is under integration in the real-time system. 

And we have re-checked and corrected English.

Reviewer 2 Report

Comments and Suggestions for Authors

The manuscript "A Hybrid System for Defect Detection on the Railways by Fusion of Object and Context Information" delineates an innovative approach for detecting railway defects through a hybrid model, amalgamating object detection with context analysis. This topic aligns with the thematic scope of the Sensors journal. Nonetheless, the manuscript could be augmented in several dimensions:

1. Ensuring initial instances of abbreviations are accompanied by their expanded forms is essential for clarity. Terms like "NNs" in the Abstract, along with "YOLO," "AI," and "SOTA" in the Introduction, and "RCNN," "DNN," and "GPU" in the Related Work, require explicit definitions upon their first appearance.

2. The manuscript's proposition to integrate a Convolutional Block Attention Module (CBAM) into the YOLOv5 architecture warrants a more comprehensive justification. Elaborating on the rationale behind this specific amalgamation and detailing its integration would substantiate the model's choice, fortifying the paper's foundational premise.

3. Amplifying the ablation studies to dissect and modify individual elements of the proposed model methodically would elucidate each component's integral role and impact. Such meticulous dissection is pivotal in comprehending the contributions of distinct facets like the attention mechanism and fusion methodology to the model's collective efficacy.

4. As currently articulated, The decision model for rule-based, decision-level fusion would benefit substantially from a more exhaustive delineation of the algorithmic processes involved. Discussing the logic underpinning the selection of specific rules and their anticipated influence on model performance would enrich the methodological discourse.

5. The discussion about image utilization for defect detection needs expansion, particularly regarding the preprocessing of these images. Furnishing details about image dimensions, resolution, and preprocessing protocols such as normalization or augmentation would clarify the quality and handling of the input data.

6. The absence of an in-depth computational complexity analysis is notable. A deeper understanding of the model's computational demands, both in time and space, especially vis-à-vis the volume of data processed, is imperative for evaluating its practical applicability.

7. The robustness of the model, considering variations in ambient conditions like lighting and weather, needs to be more exhaustively examined. This facet is paramount for the model's practical deployment in diverse environmental conditions.

Comments on the Quality of English Language

Good

Author Response

Thank you for your valued comments. We did our best to improve the paper quality accordingly. Please find below our responses:

  1. Ensuring initial instances of abbreviations are accompanied by their expanded forms is essential for clarity. Terms like "NNs" in the Abstract, along with "YOLO," "AI," and "SOTA" in the Introduction, and "RCNN," "DNN," and "GPU" in the Related Work, require explicit definitions upon their first appearance.

    We have checked and explained all the abbreviations first time they are met in the text, for instance in the abstract, see “

    “...by incorporating a Convolutional Block Attention Module (CBAM) in the You Only Look Once (YOLO) versions 5 (YOLOv5) and 8 (YOLOv8) architectures…”

    “...Deep Neural Networks (DNNs)...“  

    In Section 1 Introduction 

    “...mixing of modalities in Artificial Intelligence (AI) based...”

    “...we present the State Of The Art (SOTA) analysis...”

    In Section 2 Related work

    “...represented by Region-based Convolutional Neural Network (RCNN) series...” 

    “...speed on lower Graphics Processing Units (GPUs) compared...”

  2.  The manuscript's proposition to integrate a Convolutional Block Attention Module (CBAM) into the YOLOv5 architecture warrants a more comprehensive justification. Elaborating on the rationale behind this specific amalgamation and detailing its integration would substantiate the model's choice, fortifying the paper's foundational premise.We have explained the rational of Convolutional Block Attention Model (CBAM) in Section 3.3. , see “ Integration of attention blocks into Deep NN generally improves their performances, as channel attention and spatial attention allow for the selection of most significant feature channels and features both at training and generalization steps
  3. Amplifying the ablation studies to dissect and modify individual elements of the proposed model methodically would elucidate each component's integral role and impact. Such meticulous dissection is pivotal in comprehending the contributions of distinct facets like the attention mechanism and fusion methodology to the model's collective efficacy.We have re-organized our results and conducted supplementary analysis. The ablation studies are now presented in section 5.2 Ablation study.
  4. As currently articulated, The decision model for rule-based, decision-level fusion would benefit substantially from a more exhaustive delineation of the algorithmic processes involved. Discussing the logic underpinning the selection of specific rules and their anticipated influence on model performance would enrich the methodological discourse.

    Thank you for your remark, we have shown the underpinning logic in section 2.2. Fusion methods, see “ … Current commercial systems, using Convolutional Neural Networks (CNNs), show that image analysis by CNNs is a difficult task due to the ambiguity of the interpretation of the information contained in the images… ‘. 

    Next, in section 3.5. Rule-based decision level fusion, the domain knowledge, is presented and summarised in Table 1. Rules to be considered by the decision model to classify the different types of defects that require context.

  5. The discussion about image utilization for defect detection needs expansion, particularly regarding the preprocessing of these images. Furnishing details about image dimensions, resolution, and preprocessing protocols such as normalization or augmentation would clarify the quality and handling of the input data.

    All images are captured in normalized conditions on the embedded system with close camera view. We explain this in Section 4.1 Image datasets, see “There is no pre-processing of images. It is not needed as the images are taken from a close camera view and the system has to adapt to different conditions, mainly lightening. 

    Concerning image dimensions, we give typical dimensions and reduction rates for the adaptation of them to be processed by YOloV5 detector in the same section, see” As for image resolution, it ranges from 774 x 1480 up to 1508 x 500 as they are captured by HD cameras…”

    For the data augmentation, in the same section, please see “Data augmentation is also not necessary because the images were taken under the same conditions and the original image diversity is sufficient for a good model training.”

  6. The absence of an in-depth computational complexity analysis is notable. A deeper understanding of the model's computational demands, both in time and space, especially vis-à-vis the volume of data processed, is imperative for evaluating its practical applicability.We have put a detailed complexity analysis of different blocs of the system in section “5.5. Complexity analysis”, the information is given in Table 10. Computational complexity analysis of Yolov5 and in the underlying text.
  7. The robustness of the model, considering variations in ambient conditions like lighting and weather, needs to be more exhaustively examined. This facet is paramount for the model's practical deployment in diverse environmental conditions.                                                                            We have conducted robustness studies with regard to the domain dependent image degradations such as lightening conditions, which are basically the only degradations we can expect, as due to the normalized acquisition protocol, other degradations , such as geometrical or noise are minimal. They should not influence the performance of YOLOv5m. Thee results of these experiments and their analysis may be found now in  section 5.3. Robustness analysis. 

Reviewer 3 Report

Comments and Suggestions for Authors

The presented method is interesting, with a potential to provide a useful tool for evalutation of the state of the railway track.  The analysis is however at the early stage, currently teaching the AI to recognize deficiencies.  The observations of the context, that are given in this paper, have a potential not only to help detect a failure, but also to search for additional clues what might have caused that failure, especially that the measurements are done under a dynamic load.
English language is good, found only several minor mistakes (line in line 478, don't use The and the colon), which have no effect on the reader.  The sentences are clear and can be swiftly read.

Author Response

English language is good, found only several minor mistakes (line in line 478, don't use The and the colon),

Thank you very much for your review. Detection of railway defects is a current problem.  We have thoroughly corrected English. Application of artificial intelligence principles for its solution requires further research.

Round 2

Reviewer 2 Report

Comments and Suggestions for Authors

I have no more concerns about this paper, and it could be accepted.